# The Role of NCS1 in Immunotherapy and Prognosis of Human Cancer

**DOI:** 10.3390/biomedicines11102765

**Published:** 2023-10-12

**Authors:** Gen-Chun Wang, Xin Gan, Yun-Qian Zeng, Xin Chen, Hao Kang, Shuai-Wen Huang, Wei-Hua Hu

**Affiliations:** 1Department of Orthopedics, Tongji Hospital, Tongji Medical College, Huazhong University of Science and Technology, Wuhan 430030, China; 2Department of General Practice, Tongji Hospital, Tongji Medical College, Huazhong University of Science and Technology, Wuhan 430030, China

**Keywords:** Neural Calcium Sensor1, pan-cancer, immunotherapy, prognosis

## Abstract

The Neural Calcium Sensor1 (NCS1) is a crucial protein that binds to Ca2^+^ and is believed to play a role in regulating tumor invasion and cell proliferation. However, the role of NCS1 in immune infiltration and cancer prognosis is still unknown. Our study aimed to explore the expression profile, immune infiltration pattern, prognostic value, biological function, and potential compounds targeting NCS1 using public databases. High expression of NCS1 was detected by immune histochemical staining in LIHC (Liver hepatocellular carcinoma), BRCA (Breast invasive carcinoma), KIRC (Kidney renal clear cell carcinoma), and SKCM (Skin Cutaneous Melanoma). The expression of NCS1 in cancer was determined by TCGA (The Cancer Genome Atlas Program), GTEx (The Genotype-Tissue Expression), the Kaplan–Meier plotter, GEO (Gene Expression Omnibus), GEPIA2.0 (Gene Expression Profiling Interactive Analysis 2.0), HPA (The Human Protein Atlas), UALCAN, TIMER2.0, TISIDB, Metascape, Drugbank, chEMBL, and ICSDB databases. NCS1 has genomic mutations as well as aberrant DNA methylation in multiple cancers compared to normal tissues. Also, NCS1 was significantly different in the immune microenvironment, tumor mutational burden (TMB), microsatellite instability (MSI), and immune infiltrate-associated cells in different cancers, which could be used for the typing of immune and molecular subtypes of cancer and the presence of immune checkpoint resistance in several cancers. Univariate regression analysis, multivariate regression analysis, and gene enrichment analysis to construct prognostic models revealed that NCS1 is involved in immune regulation and can be used as a prognostic biomarker for SKCM, LIHC, BRCA, COAD, and KIRC. These results provide clues from a bioinformatic perspective and highlight the importance of NCS1 in a variety of cancers.

## 1. Introduction

The main treatment for advanced cancer is immunotherapy, but there are many patients who do not respond to it, limiting its clinical applicability. Therefore, the study of new immune markers and understanding the mechanisms of immune marker resistance is a research priority [1]. 

Intracellular calcium signaling plays a role in regulating cancer progression. Ca2^+^-binding proteins can change their conformation upon binding Ca2^+^ and bind to other effector proteins, which is crucial for regulating cellular homeostasis [2]. Neuronal Calcium Sensor 1 (NCS1) is a calcium-binding protein expressed ubiquitously. It is involved in encoding calcium-binding proteins that regulate various processes in the cellular life cycle, promoting cell survival and tumor invasion [3,4,5]. Multiple transcript variants encoding different isoforms have been identified for this gene [6]. NCS1 has been suggested to enhance the function and persistence of chimeric antigen receptor T-cell immunotherapy (CAR-T) cells via TH17 cells in hematologic malignancies [7]. It promotes the proliferation and differentiation of ovarian granulosa cells [8,9], reduces survival in breast cancer, promotes malignancy in squamous lung cancer, and enhances invasion and metastasis in prostate, breast, and glioma cancers [10,11,12,13].

NCS1 is a crucial calcium-binding protein believed to play a role in regulating tumor invasion and cell proliferation. However, its role in immune infiltration and cancer prognosis is still unknown. Tumor immune escape mechanisms exist during cancer development, involving the immune checkpoint blockade (ICB) pathway. Tumor mutational burden (TMB) is used to determine tumor antigenicity. Deficient mismatch repair (dMMR) and microsatellite instability (MSI) can promote the accumulation of somatic mutations in tumor cells, increasing TMB and the susceptibility to ICB [14]. Understanding tumor genetics and immune and non-immune factors in the tumor microenvironment can facilitate the development of new diagnostic, prognostic, and therapeutic strategies to improve the prognosis of cancer patients [15].

This study employed machine learning, transcriptome mapping from The Cancer Genome Atlas (TCGA), and relational databases to assess the enriched expression pattern of NCS1 in different cancer subtypes. It also investigated the impact of NCS1 on immune response and prognosis. The study explored the correlation between NCS1 expression and tumor-infiltrating lymphocytes (TILs), as well as associated immune markers. The researchers delved into the intricate and distinctive relationship between NCS1 and tumors. Moreover, they demonstrated that high NCS1 expression in cancer patients reduces their survival time and further confirmed the pro-cancer role of NCS1 through biological validation using immunohistochemical staining.

## 2. Materials and Methods

### 2.1. Data Collection

TCGA tumor type and GTEx (The Genotype-Tissue Expression) normal-tissue-type data were collected. The data sources were RNAseq data from TCGA [16] and UCSC XENA uniformly processed GTEx data [17]; the data were normalized using the “EdgeR” R package; RNAseq data were converted in transcription per million reads (TPM) format and log2 (TPM+1) transformation; the dataset was filtered to retain clinical information and remove the results of missing and duplicate samples. TMB was determined by counting the number of insertion or deletion events in duplicate gene sequences, while MSI was determined by counting the overall mutation incidence per million base pairs [17].

### 2.2. Evaluate NCS1 Expression Level and the Degree of DNA Methylation

To determine the potential prognostic value and correlation between NCS1 and clinicopathological features of the tumor, several tools and databases were utilized. These include GEPIA 2.0 [18], Kaplan–Meier Plotter [19,20], The Human Protein Atlas [21], and tissue microarray. GEPIA 2.0 is a tool that allows the analysis of RNA sequencing expression data from TCGA and GTEx programs, enabling the investigation of the association between NCS1 expression and cancer prognosis, including its correlation with the pathological stage. The “Survival” module of GEPIA 2.0 is particularly useful for exploring these relationships [18]. The Human Protein Atlas provides protein expression profiles for a large number of genes in various tissues and organs. The tissue microarray (TMA) used in this study, MTU1021, was purchased from Guilin Yao Li Biotechnology Co., Ltd. It contains samples from 28 organs and 41 tumor types, with each tumor surrounded by adjacent organ tissues. NCS1 expression was detected using the NCS1 rabbit monoclonal antibody (AB129166, Abcam, Cambridge, UK).

Additionally, the UALCAN database was used to evaluate NCS1 DNA promoter region methylation in pan-cancer data [22], which includes data from TCGA, met500, and Clinical Proteomic Tumor Analysis Consortium, to determine the epigenetic regulation of NCS1.

### 2.3. Evaluate NCS1 and Immune Infiltration

T cell immune estimation resource (TIMER 2.0) (http://timer.comp-genomics.org/, accessed on 19 September 2021) uses the deconvolution method to estimate the abundance of tumor-infiltrating immune cells (B cells, CD4+T cells, CD8+T cells, macrophages, neutrophils, and dendritic cells) [23]. A comparison of tumor infiltration between tumors with different somatic copy number variants of NCS1 was analyzed by the “SCNA” module of TIMER 2.0 [23]. Based on multiple tumor RNAseq data and relevant clinical information obtained from the TCGA database, to evaluate reliable results for immune score assessment, we used the “immuneeconv” package. This R package integrates six cutting-edge algorithms: TIMER, xCell, MCP-counter, CIBERSORT, EPIC, and stromal score. It derives the immune score, tumor microenvironment score, and stromal score for NCS1 in pan-cancer, enabling the assessment of NCS1’s immune relevance in the context of pan-cancer. Over 40 widely recognized immune checkpoint genes were gathered, and their expression values were extracted to investigate the expression patterns of immune checkpoint-related genes in relation to NCS1 across various types of cancer.

### 2.4. Association between NCS1 and Immunotherapy

Our comprehensive immunogenicity analysis of NCS1 was conducted using the R software package(4.1.2), the cBioPortal [24], and the TISIDB database [25]. There are two main sources of tumor immunogenicity differences: internal tumor factors and external tumor factors. Internal tumor factors include neoantigen frequency, tumor mutation load, immune inhibitors, immune-stimulators, and major histocompatibility complex (MHC) molecules. Among the external tumor factors are tumor-infiltrating lymphocytes (TILs) and chemokines (or receptors for chemokines) that regulate T-cell transport [25]. The R package and the TISIDB database were used to analyze the TMB and MSI of NCS1 in pan-cancer. The TISIDB database modules “Immunomodulators”, “Chemokines”, and “Lymphocytes” were used to analyze the association between NCS1 and immunosuppressive, immunostimulatory, and MHC molecules, as well as TILs and chemokines regulating T-cell trafficking in pan-cancer. Spearman correlation analysis was performed to assess whether NCS1 regulates immune function in cancer. The “Subtypes” module of the TISIDB database was used to explore the distribution of NCS1 expression in molecular and immune subtypes of human cancers. Clinical features related to NCS1 such as pathological staging and grading were explored in human cancers. The “Drugs” module was used to identify relevant drugs for NCS1, which can help design therapeutic approaches in combination with immunotherapy.

The Tumor Immune Dysfunction and Exclusion (TIDE) algorithm from tide.dfci.harvard.edu [26] combines the expression profiles of T-cell dysfunction and T-cell rejection to model tumor immune escape based on TIDE scores. It can be used to predict immune checkpoint blockade (ICB) resistance and identify biomarkers of ICB response.

### 2.5. Relationship between NCS1 and Prognosis of Tumor Patients 

This study conducted both univariate and multifactorial Cox regression analyses to investigate the association between NCS1 and tumor clinical characteristics, specifically tumor grade and tumor stage. The findings were presented using forest plots generated with the “forest plot” package, which displayed each factor’s *p* value, hazard ratio (HR), and 95% hazard ratio (CI). Based on the results of the multifactorial Cox proportional risk analysis, column plots were created using the “RMS” package to predict the overall recurrence rate at 1, 3, and 5 years. The line graphs visually depict these factors and enable the calculation of individual patient prognostic risk based on the points associated with each risk factor. In the column line diagram, line segments represent the range of values for each variable, with the length indicating the magnitude of the factor’s impact on the outcome event. Individual scores are represented by points on the diagram, reflecting the scores corresponding to each variable at different values. The Total Point represents the cumulative score derived from all variables. By summing the scores of all clinical indicators, the total score is obtained, which can be used to infer the patient’s future survival rate at 1, 3, and 5 years.

NCS1 was identified as a variable in the Nomogram based on the results of both univariate and multifactorial Cox regression analyses. If NCS1 shows significant differences in prognosis in both analyses, it suggests that the gene is an independent variable regardless of other clinical factors. The nomogram’s discrimination was evaluated by plotting observed rates against nomogram-predicted probabilities, and the concordance index (C-index) was calculated using a bootstrap method. A closer alignment between the nomogram model and the calibration curve indicates better predictive accuracy.

To analyze the impact on the 1-, 3-, and 5-year survival rates of tumor patients, the “pROC” package was utilized for analysis and the “ggplot2” package for visualization. Bootstrap correction and 200 replicate bootstrap samples were employed to assess the internal validity of the constructed comparison plots. Kaplan–Meier curves were generated using the “RMS”, “Survival”, and “Survminer” packages, providing *p*-values and HR with 95% CI for analysis via log-rank tests.

### 2.6. NCS1-Related Gene Enrichment Analysis

The 100 most relevant genes for NCS1 were obtained from GEPIA2.0 (Appendix A) and the list of genes was analyzed using Metascape (https://metascape.org, accessed on 19 September 2021) [27] for GO/KEGG, transcription factors, PaGenBase, DisGeNET, and Coronavirus disease (COVID) analysis [27,28]. PaGenBase is used to understand the functions of NCS1-like genes and their roles in specific biological processes and promote drug development and disease mechanism [29]. DisGeNET is used for a variety of research purposes, including analyzing the molecular basis of human diseases, verifying predicted disease-causing genes, and evaluating text mining performance [30]. A COVID analysis was performed to confirm that NCS1 is a reliable marker for COVID, followed by a principal components analysis (PCA) to determine whether NCS1 distributions in para-cancerous and tumorous tissues were correlated [31]. NCS1 particular knockout cell lines were subjected to ICSDB (https://icsdb.lk, accessed on 19 September 2021) to evaluate the changes in cell proliferation and to verify its efficiency [32].

### 2.7. The Structure of NCS1 and Screening of Therapeutic Drugs

To explore the clinical application of NCS1 and the value of transformation, we searched the pharmacogenetics Drugbank (https://go.drugbank.com, accessed on 19 September 2021) [33], pharmGKB (https://www.pharmgkb.org, accessed on 19 September 2021) [34], and chEMBL databases (https://www.ebi.ac.uk/chembl, accessed on 19 September 2021) [35] for the compound structure.

### 2.8. Statistical Analysis

Spearman correlation analysis was employed to evaluate the correlation between continuous variables. Paired *t* tests or *t* tests were utilized to compare the NCS1 expression levels between groups or between tumors and normal tissues, depending on whether the samples were paired. Kaplan–Meier survival curves were generated to assess the survival probabilities. Cox regression analysis was conducted using the R survival data package, with a significance level set at *p* < 0.05. Graphs were created using R packages such as “forest plot”, “RMS”, “ggplot2”, “pROC”, “Survival”, and “Survminer”. Two-tailed Student’s *t* tests were used to compare differences between two groups, while the Wilcoxon test was employed for two-group data. One-way ANOVA was performed to determine statistical significance among more than two groups. Mean ± SD values were reported for data. Statistical significance was defined as *p* < 0.05. All statistical analyses were conducted using SPSS (version 24.0.0).

## 3. Results

### 3.1. The Landscape of NCS1 Status in Pan-Cancer 

The expression of NCS1 in 33 human cancers was determined using the TCGA and GTEx portals. NCS1 was found to have significant differences in different tumor stages of pan-cancer (Figure 1A). Kaplan–Meier survival analysis showed that the high expression of NCS1 predicted low levels of overall (OS) and disease-free survival (DFS) (*p* < 0.01) (Figure 1B,C). Analysis of RNAseq data in TPM format from UCSC Xena (https://xenabrowser.net/datapages/, accessed on 19 September 2021) processed uniformly by the Toil process [16] for TCGA and GTEx showed that the difference in NCS1 expression was significant in 27 of the 33 tumor types. NCS1 was not expressed in CESC (Cervical Cancer), LUAD (Lung Adenocarcinoma), PCPG (Pheochromocytoma & Paraganglioma), and in MESO (Mesothelioma), SARC (Sarcoma), and UVM (Ocular melanomas) due to some missing data, and the results could not be derived (Figure 1D, Appendix A). HPA immunohistochemistry data showed that NCS1 expression in BRCA and LIHC was significantly different from that in normal tissues (Figure 1E). According to the TMA data, the high expression of NCS1 was observed in KIRC (Kidney Clear Cell Carcinoma), SKCM (Melanoma), and BRCA (Breast Cancer) (Figure 1F).

### 3.2. The Degree of DNA Methylation and Tumor–Immune Interaction of NCS1 in Pan-Cancer

The outcomes indicated that NCS1 had hypermethylation in LIHC (Liver Cancer) (*p* value = 1.09 × 10^−8^), BRCA (*p* value = 1.09 × 10^−8^), COAD (Colon Cancer) (*p* value = 5.69 × 10^−8^), and KIRC (*p* value < 1 × 10^−12^) (Figure 2A). Tumor heterogeneity is a tumor-specific factor that should be considered when searching for molecular markers [36,37]. We examined the association between ncs1 and the degree of immune cell infiltration in multiple tumor types by using the Tumor Immune Assessment Resource (Timer) database, and the correlation linear regression plots showed a correlation between the high expression of NCS1 and increased immune cell infiltration in BRCA, LIHC, KIRC, and COAD. Notably, in COAD, NCS1 showed a negative correlation with B cells. There was a significant correlation with increased tumor invasion by antigen-presenting cells (APCs) (Figure 2B). APCs may directly process NCS1. T cells may present it to B cells, while T cells may recognize and activate it. Molecular characterization based on the somatic mutation profiles has been shown to detect a correlation between mutations and cancer prognosis [38,39,40]. To gain a deeper understanding of the association between NCS1 genomic indicators and immune infiltration across various cancer types, SCNAs were categorized into five levels: deep deletion, arm-level deletion, normal, arm-level gain, and high amplification. The findings revealed significant variations in immune cell enrichment among different NCS1 SCNAs in pan-cancer, indicating varying degrees of immune cell infiltration (Figure 2C,D). The R package integrated six latest algorithms, including TIMER, xCell, MCP-counter, CIBERSORT, EPIC, and quanTIseq [41]. Significant differences were obtained between NCS1 expression and pan-cancer in Stromal Score, tumor microenvironment score, immune score, and immune infiltration-related cells (Figure 2E). It is suggested that the genomic alteration of NCS1 in pan-cancer is closely related to the degree of immune infiltration.

### 3.3. Correlation between NCS1 and Tumor Immunogenicity in Pan-Cancer

Immune checkpoint genes represent novel targets for the development of cancer therapies, while there is a significant correlation between the expression pattern of immune checkpoint genes and patient survival and response to ICB therapy [42]. We collected over 40 common immune checkpoint genes and concluded that a significant relationship exists between immune checkpoint genes (CD276, CD274, NRP1, LAG3, LAIR1, LGALS9, and VSIR) and NCS1. 

In LUAD, LUSC (Lung Squamous Cell Carcinoma), PRAD (Prostate Cancer), UCEC (Endometrioid Cancer), BLCA (Bladder Cancer), TGCT (Testicular Cancer), ESCA (Esophageal Cancer), LIHC, CESC, SARC (Sarcoma), BRCA, COAD, SKCM, CHOL (Bile Duct Cancer), KIRC, THCA (Thyroid Cancer), HNSC (Head and Neck Cancer), LGG, KICH (Kidney Chromophobe), and UVM, the expression of NCS1 showed a positive correlation with immune checkpoint genes, especially in TGCT, LIHC, BRCA, SKCM, COAD, KICH, and UVM. It is suggested that NCS1 may be similar to the above genes or have some standard biological processes involved in coordinating the activity of immune checkpoint genes in different signaling pathways, which may lead to reduced survival and a diminished response to immune checkpoint blockade therapy (Figure 3A).

The impact of NCS1 on the tumor microenvironment (TME) immune mechanisms and immune response was investigated by analyzing the correlation between NCS1 expression and tumor mutation burden (TMB) as well as microsatellite instability (MSI) [43]. Tumor mutation burden (TMB) is a measure that quantifies the number of mutations present in cancer cells [43,44,45]. The researchers calculated the tumor mutation burden (TMB) for each tumor sample and examined its association with NCS1 using Spearman’s rank correlation coefficient. The results indicated a positive correlation between NCS1 expression and TMB in UCEC and SKCM (Figure 3B). Microsatellites (MS) are short sequences of repeated DNA in the human genome that play an important role in tumor development. Microsatellite instability (MSI) is considered a molecular marker for prognosis of colorectal cancer. NCS1 was positively correlated with microsatellite instability in BRCA, DLBC, UCS, ACC (Adrenocortical Cancer), LUSC, TGCT, and LIHC (Figure 3C) and negatively correlated with expression in KICH, THCA, and STAD (Figure 3C). The researchers visualized and scored immunophenotypes, including tumor-infiltrating lymphocytes (TILs) (Figure 3D), immune-inhibitors (Figure 3E), immune-stimulators (Figure 3F), MHC molecules (Figure 3G), chemokines (Figure 3H), and chemokine receptors (Figure 3I). A genetic screening technique based on the ICSDB database was used to investigate whether the NCS1 gene could increase T cell-mediated sensitivity and tumor cell resistance by interacting with 28 types of TILs. According to the multivariate analysis, the NCS1 immunophenotype scores varied among different tumors.

NCS1 is believed to be associated with the molecular and immune subtypes of tumors (Figure 4A,B). Using TCGA immune-related gene expression data as a training set, NCS1 could be used to predict the immune subtypes of LIHC (*p* = 1.91 × 10^−4^), BRCA (*p* = 1.94 × 10^−26^), COAD (*p* = 4.47 × 10^−3^), and KIRC (*p* = 2.64 × 10^−5^) (Figure 4C). Based on the molecular subtype classification from the TCGA database, NCS1 was a good predictor of molecular subtypes including SKCM (*p*-value = 4.51 × 10^−2^), LIHC (*p*-value = 3.36 × 10^−9^), BRCA (*p*-value = 6.25 × 10^−64^), and COAD (*p*-value = 3.09 × 10^−2^) (Figure 4D).

The TIDE algorithm was utilized to investigate the role of NCS1 in immunotherapy. High TIDE scores were associated with elevated expression of NCS1 in LIHC, BRCA, and KIRC; SKCM exhibited low NCS1 expression, resulting in a more favorable response to immune checkpoint blockade (ICB) therapy (Figure 5A). As a regulator of immune escape in tumors, elevated expression of NCS1 in cancer cells leads to resistance against T-cell-mediated killing effects during immune checkpoint blockade (ICB) therapy (Figure 4E). 

### 3.4. There was a Significant Correlation between NCS1 Expression and Survival Rate

Patients were divided into high/low expression groups based on the median expression of NCS1. Univariate and multivariate COX regression analyses were conducted to identify significant prognostic variables and to create a line graph (nomogram) for prognostic guidance. A prognostic line graph for cancer was developed based on the results of the multivariate Cox proportional hazards analysis. Univariate regression analysis identified NCS1, age, sex, race, and pTNM-stage as predictors related to overall survival (OS) in SKCM, LIHC, and KIRC. In the multifactor regression analysis, after adjusting for clinical characteristics such as age and sex, NCS1 was associated with SKCM (HR = 1.22125, *p*-value = 0.00403) and LIHC (HR = 1.26497, *p*-value = 0.00239). It was concluded that NCS1 could be an independent predictor of survival outcome in SKCM and LIHC, confirming the independent prediction of NCS1 in SKCM and LIHC survival prognosis stability. A prognostic model was constructed using multivariate Cox analysis to generate ROC curves for predicting the 1-, 3-, and 5-year survival of patients with SKCM, LIHC, and KIRC, and the corresponding AUC was calculated. The results showed that the 1-, 3-, and 5-year prediction curves in the constructed prognostic model were in good agreement with the actual results. In addition, KM curves were constructed to observe the relationship between NCS1 and overall survival (OS), with high expression of NCS1 in SKCM (HR = 1.53, P = 0.002), LIHC (HR = 1.50, P = 0.02), BRCA (HR = 1.40, P = 0.038), COAD (HR = 1.51, P = 0.041), KIRC (HR = 1.49, P = 0.041), and KIRC (HR = 1.49, P = 0.0009), which was associated with a decrease in OS (Figure 5B). Additionally, the expression levels of NCS1 significantly varied among different pathological stage and grade groups in KIRC tumors, as well as in different pathological stage groups in BRCA tumors (Figure 5C,D).

### 3.5. GO/KEGG Enrichment Analysis of NCS1 and NCS1 Similar Genes

GEPIA 2.0 identified 100 genes that exhibited the highest similarity to NCS1 in terms of biological function, and these genes were analyzed using Metascape [27]. The results revealed that these genes were enriched in Gene Ontology (GO) terms related to chemical synaptic transmission, synaptic vesicle cycle, learning, and memory (Appendix A), with a concentration of biological processes related to signaling, localization, and behavior (Appendix A). A protein–protein interaction network of genes similar to NCS1 was presented [46] (Appendix A). The MCODE algorithm was utilized to identify densely connected protein neighborhoods and predict the most significant hub genes and pathways within the network [47]. GO enrichment analysis was conducted on each MCODE network to assign functional annotations to the network components. The results show that the essential genes are CAMK2A, SHANK1, DLG4, CACNG3, CACNG8, CNIH2, SYN1, SYN2, RAB3A, SEPTIN3, SK1SEPTIN5, GNAO1, GNG3, and PRKCG (Appendix A); the main biological processes are protruding through Chemical Transmission, Antegrade Trans-synaptic Signaling, and Trans-synaptic Signaling (Appendix A). Gene list enrichment was observed in the following ontology categories: Transcription Factor Targets [48], DisGeNET [30], PaGenBase [29], and COVID [27]. Based on these findings, EFCQ6 and RFX102 were identified as targets of a transcription factor that regulates NCS1-like genes (Figure 6A).

Based on DisGeNET, NCS1 and its related genes are primarily associated with phenotypes of Lewy body disease (Figure 6B). Immunosuppression and immune dysfunction weaken the immune system in cancer patients. Peripheral blood mononuclear cells (PBMCs) play a vital role in hematological malignancies, vaccine development, and immune disorders, particularly in response to tumor antigens. Furthermore, COVID-19 patients have a higher likelihood of developing colorectal cancer, lung cancer, and breast cancer compared to non-cancer patients [49]. Based on the aforementioned reasons, there is a significant difference in PBMCs between COVID-19 patients and non-COVID-19 patients regarding NCS1 and its related genes (Figure 6C), indicating that NCS1 could serve as a potential therapeutic target for COVID-19, and the development of an NCS1 mRNA vaccine is plausible. PaGenBase offers a pathway for the development of novel drugs. In conclusion, NCS1 and its related genes are clearly expressed in amygdala, cortex, and DRG cell lines (Figure 6D), providing a choice of experimental direction for further NCS1 research. Principal component analysis (PCA) can be employed to differentiate tumors from normal tissues based on NCS1 and related genes (Figure 6E,F).

We conducted a screening of 976 cell lines from the ICSDB database to assess proliferation following CRISPR knockout of NCS1. The knockout of NCS1 resulted in impaired viability in cell lines such as LNCAP (PRAD, z-score = −2.163, *p* = 0.015), PANC1 (PAAD, z-score = −2.163, *p* = 0.015), G564NS (GBM, z-score = −1.649, *p* = 0.050), and BJ (skin, z-score = −1.690, *p* = 0.046). Conversely, RERFLCAI (LUSC, Z-score = 1.672, *p* = 0.047) and SNU1544 (COAD, Z-score = 1.732, *p* = 0.042) showed increased viability (Figure 7A); We created a visualization of the guide RNA-level screening to demonstrate the knockout efficiency of each gRNA along with the background distribution (Figure 7B,C). We retrieved drugs targeting NCS1 from ChEMBL (Figure 7D) and DrugBank (Figure 7E,F). Furthermore, we identified other targets of DB110933 and DB11348 that might also play a role in cancer development.

## 4. Discussion

Research has demonstrated that dysregulation of the Ca2^+^ pathway plays a role in promoting cancer invasion and metastasis. Molecules that bind Ca2^+^ and mediate the expression of downstream effector molecules may serve as potential therapeutic targets for cancer treatment [50]. NCS1 overexpression has been shown to enhance invasiveness in breast and lung squamous cancers. Alterations in the expression levels of key components or complexes involved in cellular Ca2^+^ transduction can lead to malignant transformation, tumor progression, abnormal tumor cell proliferation, and resistance to cell death. These factors also play a crucial role in immunotherapy sensitivity [51]. The dysregulation of the Ca2^+^ pathway contributes to the promotion of cancer invasion and metastasis. Molecules that bind Ca2^+^ and mediate the expression of downstream effector molecules may hold promise as potential therapeutic targets for cancer treatment. Research has demonstrated that NCS1 overexpression promotes invasion and proliferation in various cancers, such as prostate, breast, glioma, ovarian, and hematologic malignancies [52].

Given that NCS1 plays a vital role in regulating immune tolerance and tumor development across various histological types in pan-cancer, we thought it might be routinely applied to various tumors. The findings revealed that high expression levels of NCS1 were associated with reduced overall survival (OS) and disease-free survival (DFS), indicating significant tumor heterogeneity. To validate the expression of NCS1 in pan-cancer at the histological level, the researchers used TMA microarray and HPA data, which showed significant differences in NCS1 in LIHC, BRCA, KIRC, and SKCM compared to normal tissues. Genetically unstable cancers exhibit increased genetic diversity and mutational load, leading to the generation of novel antigens and enhanced immune infiltration, resulting in amplification, profound deletions, missense mutations, and other genetic changes in several types of cancers. As per the findings of the ENCODE project, protein heterodimers are thought to be prevalent in important cancer genes [53]. Switching of heterodimers promotes hepatocellular carcinoma cell proliferation and tumor formation [54,55], while variable splice switching of cancer genes uncovers novel characteristics of cancer that exhibit strong predictive power for patient survival [56,57]. The discovery that NCS1 can reduce cross reactivity with other proteins and exhibits tumor-specific protein isoforms has stimulated fundamental mechanistic studies focused on NCS1.

Pan-cancer differentially methylated CpG sites (PDMCs) in tumors play a role in regulating tumor suppressor genes or oncogenes, contributing to tumor growth and impacting patient survival [58]. DNA methylation can alter the chromatin structure to maintain the balance between transcriptional activation and repression. The same gene can exhibit distinct DNA methylation patterns across different tumors, which correlates with its gene expression pattern [58]. NCS1 demonstrates differential methylation across at least eight tumor types, making it a pan-cancer-wide differentially methylated CpG site capable of delineating distinct functional groups. These functional groups are implicated in various tumor-related signaling pathways [59]. We examined the correlation between NCS1 methylation levels and overall survival in patients with different cancer types, revealing its significant prognostic value in more than four cancer types. This study lays the foundation for further research on the impact of NCS1 methylation on cancer. The objective of tumor immunotherapy is to reactivate and sustain the tumor–immune cycle, restoring a normal antitumor immune response in the body to effectively control and eliminate tumors. By considering the cellular characteristics of immune infiltration, tumor genotype, immune phenotype, and tumor escape mechanisms, one can determine the interplay between these factors. Different tumor genotypes have different patterns of immune infiltrating cells. T cells are considered to be an important institution responsible for the antitumor effect in immunotherapy. Tumor-infiltrating T cells can be used to factor the patient’s immune background into the tumor prognosis, which is of interest. It is important to note that the immune score does not exclude other screening indicators, which can help to improve the accuracy of patient prognosis assessment. Understanding and acknowledging the immune background can potentially alter the approach to tumor treatment, leading to a more comprehensive therapeutic strategy [60]. 

Our findings revealed significant differences in immune infiltration-associated cells with respect to NCS1 expression across different tumor types. These differences were influenced by specific genomic mutations that impacted the level of immune cell enrichment. Furthermore, the stromal score, tumor microenvironment score, and immune score exhibited distinct expression patterns of NCS1 across various tumor types. To comprehend the impact of NCS1 on the immune system, it is essential to consider immune-related gene ontologies, encompassing immune checkpoint gene expression, tumor mutation burden (TMB), microsatellite instability (MSI), T cell activation, T cell proliferation, T cell differentiation, chemokines, and B cell receptor signaling pathways. In addition, immune infiltration is associated with the expression levels of immunosuppressants, immune agonists, and MHC molecules. Tumor-infiltrating lymphocytes (TILs) have been identified as independent predictors of tumor prognosis and immunotherapeutic efficacy [61,62]. Studies have demonstrated correlations between the levels of NCS1 expression and the numbers of TILs, immunosuppressants, immunostimulants, MHC molecules, chemokines, and chemokine receptors in different tumor types. Additionally, co-expression of NCS1 and immune checkpoint genes, which exhibit significant variations in immune infiltration and immunogenicity, provides a theoretical foundation for combined molecular targeted immunotherapy. Tumors exhibit diverse immune subtypes and molecular subtypes, leading to variations in the expression and prognosis of immune-regulated genes within these subtypes. These differences offer new perspectives for tumor treatment [63]. NCS1 is involved in immune regulation and has the potential to become a pan-cancer diagnostic biomarker, exhibiting significant differences in various immune subtypes and molecular subtypes of tumors. To understand the relationship between NCS1 and immunogenicity, as well as immune infiltration, researchers utilized a precise immuno-oncology framework. The TIDE database was employed to determine how NCS1 responds to immune checkpoint inhibitors. Due to its high expression in cancer cells, NCS1 demonstrates a resistance profile against T cell dysfunction and immunosuppressive therapies, which can promote resistance to the killing effect induced by T cells during ICB treatment. This reveals that NCS1 may function as a modulator of tumor immune escape and contribute to ICB resistance. After confirming the role of NCS1 in immunotherapy, univariate and multivariate COX regression analyses were conducted to construct KM curves, further determining the effect of NCS1 on the survival of tumor patients. NCS1 showed a negative correlation with the prognosis of SKCM, LIHC, BRCA, COAD, and KIRC and could be used as an independent prognostic factor in SKCM, LIHC, and KIRC. It exhibits good predictive accuracy and correlates with clinical features of various tumors. Pathway enrichment and principal component analysis of NCS1-like genes were performed to elucidate the mechanism of NCS1 action. Additionally, NCS1 is considered a potential biomarker for COVID and could be utilized in exploring therapeutic immune vaccine studies based on NCS1 mRNA neoantigens [64]. According to the ICSDB database, NCS1 knockdown after CRISPR inhibited the proliferation of PRAD, PAAD, and GBM cell lines, providing a reference for subsequent cytological experiments to test the effect of NCS1 knockdown on tumor cell proliferation. Evidence suggests that NCS1 mutations can lead to drug resistance [65]. 

### Limitations of the Study

Firstly, it is important to acknowledge potential data bias due to racial factors when expanding the study, as the databases used for analysis and processing are derived from publicly available sources. Secondly, to gain a deeper understanding of the impact of NCS1, it would be beneficial to explore its effects on the proliferation and invasion of various tumor cell lines at the cellular and histological levels, in addition to detecting NCS1 expression levels solely through immunohistochemical staining of tumor tissues. Lastly, conducting clinical drug trials is necessary to evaluate the efficacy of potential compounds targeting NCS1.

## 5. Conclusions

In summary, our investigation encompassed NCS1 expression patterns, correlation with tumor immune infiltration and immune infiltrating cells, prognostic value, enrichment pathways, and potential targeting drugs from multiple perspectives. Histological validation using tissue microarrays (TMAs) confirmed the association of NCS1 expression with poor prognosis in patients with SKCM. Our study sheds light on the potential value of NCS1 in cancer prognosis and immunotherapy and contributes to a deeper understanding of the mechanisms underlying NCS1’s interaction with various immune cells in tumors.

## Figures and Tables

**Figure 1 biomedicines-11-02765-f001:**
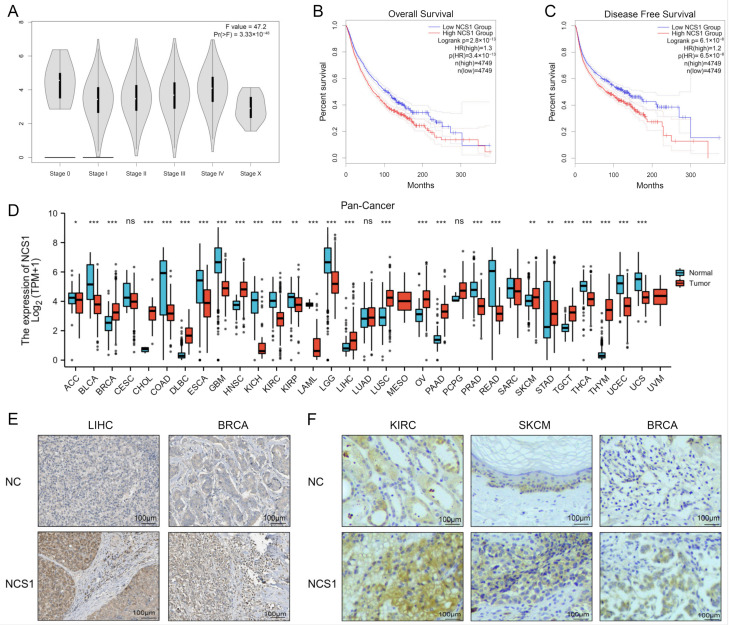
The landscape of NCS1 status in pan-cancer. (**A**) The correlation between NCS1 expression and the pathological stage of cancer using “pathological stage staging” of GEPIA2.0 found that NCS1 was significantly correlated with the pathological stage of tumors in pan-cancer. (**B**,**C**) The Kaplan–Meier plotter database was used to analyze the survival curves of high and low NCS1 expression in 33 human cancer types and determined that NCS1 was associated with lower levels of OS and DFS. (**D**) The combined data of TCGA and GTEx show the difference in the NCS1 expression in tumor and normal tissues of 33 cancers. * *p* < 0.05, ** *p* < 0.01, *** *p* < 0.001. (**E**) Representative IHC images of NCS1 expression in normal liver tissue, breast tissue, and LIHC, BRCA tissue from HPA immunohistochemistry data. (**F**) Representative IHC of NCS1 expression in normal kidney, skin, and breast tissues with corresponding KIRC, SKCM, and BRCA tissues obtained according to TMA MTU1021 images. Abbreviation: ACC: Adrenocortical Cancer; BLCA: Bladder Cancer; BRCA: Breast Cancer; CESC: Cervical Cancer; CHOL: Bile Duct Cancer; COAD: Colon Cancer; DLBC: Large B-cell Lymphoma; ESCA: Esophageal Cancer; GBM: Glioblastoma; HNSC: Head and Neck Cancer; KICH: Kidney Chromophobe; KIRC: Kidney Clear Cell Carcinoma; KIRP: Kidney Papillary Cell Carcinoma; LAML: Acute Myeloid Leukemia; LGG: Lower Grade Glioma; LIHC: Liver Cancer; LUAD: Lung Adenocarcinoma; LUSC: Lung Squamous Cell Carcinoma; MESO: Mesothelioma; OV: Ovarian Cancer; PAAD: Pancreatic Cancer; PCPG: Pheochromocytoma & Paraganglioma; PRAD: Prostate Cancer; READ: Rectal Cancer; SARC: Sarcoma; SKCM: Melanoma; STAD: Stomach Cancer; TGCT: Testicular Cancer; THCA: Thyroid Cancer; THYM: Thymoma; UCEC: Endometrioid Cancer; UCS: Uterine Carcinosarcoma; UVM: Ocular melanomas.

**Figure 2 biomedicines-11-02765-f002:**
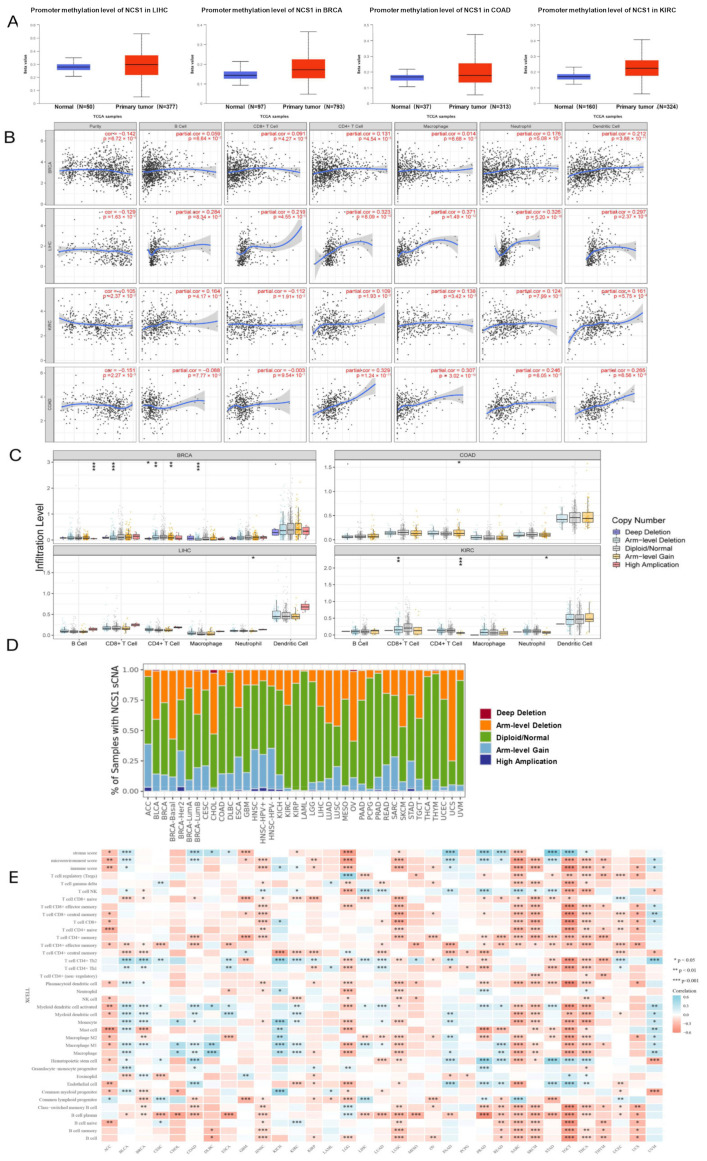
Association of NCS1 expression with DNA methylation, genomic mutations, and immune infiltrating associated cells in pan-cancer; (**A**) DNA methylation of NCS1 in LIHC, BRCA, COAD, and KIRC. (**B**) Association of NCS1 with tumor-infiltrating immune cells in BRCA, LIHC, KIRC, and COAD. (**C**) Association of NCS1 genomic indicators with immune infiltration in BRCA, LIHC, KIRC, and COAD, * *p* < 0.05, ** *p* < 0.01, *** *p* < 0.001. (**D**) Relationship between the SCNA of NCS1 and immune infiltration in different cancer types. (**E**) Heat map of correlation between NCS1 and mesenchymal score, microenvironmental score, immune score, and immune infiltrate-associated cells in pan-cancer. Abbreviation: ACC: Adrenocortical Cancer; BLCA: Bladder Cancer; BRCA: Breast Cancer; CESC: Cervical Cancer; CHOL: Bile Duct Cancer; COAD: Colon Cancer; DLBC: Large B-cell Lymphoma; ESCA: Esophageal Cancer; GBM: Glioblastoma; HNSC: Head and Neck Cancer; KICH: Kidney Chromophobe; KIRC: Kidney Clear Cell Carcinoma; KIRP: Kidney Papillary Cell Carcinoma; LAML: Acute Myeloid Leukemia; LGG: Lower Grade Glioma; LIHC: Liver Cancer; LUAD: Lung Adenocarcinoma; LUSC: Lung Squamous Cell Carcinoma; MESO: Mesothelioma; OV: Ovarian Cancer; PAAD: Pancreatic Cancer; PCPG: Pheochromocytoma & Paraganglioma; PRAD: Prostate Cancer; READ: Rectal Cancer; SARC: Sarcoma; SKCM: Melanoma; STAD: Stomach Cancer; TGCT: Testicular Cancer; THCA: Thyroid Cancer; THYM: Thymoma; UCEC: Endometrioid Cancer; UCS: Uterine Carcinosarcoma; UVM: Ocular melanomas.

**Figure 3 biomedicines-11-02765-f003:**
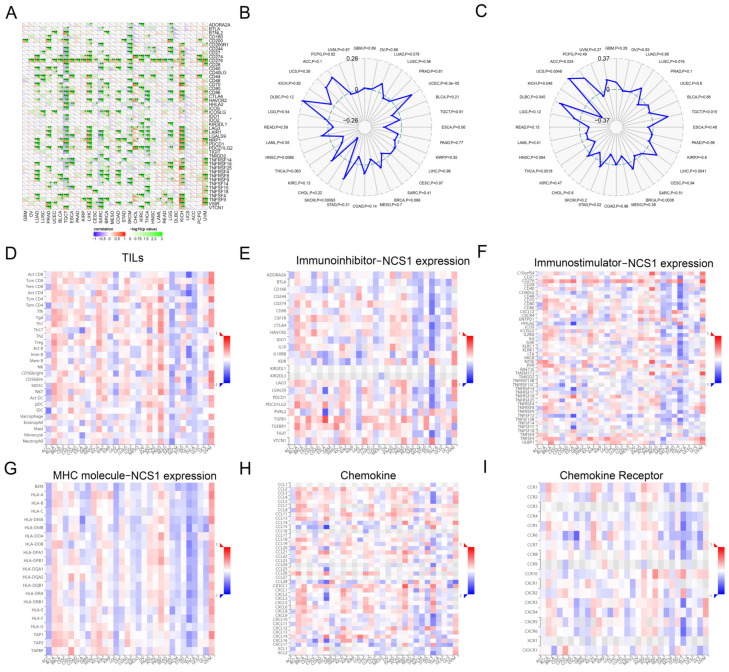
Correlation analysis between NCS1 and the immune checkpoint genes, TMB, MSI, and immune infiltration. (**A**) NCS1 expression correlates with immune checkpoint genes. * Represents *p* < 0.05, ** *p* < 0.01, *** *p* < 0.001.(**B**) Relationship between NCS1 expression and TMB. (**C**) Relationship between NCS1 expression and microsatellite instability. (**D**–**I**) Relationship between NCS1 expression and TILs, immunosuppressants, immunostimulants, MHC molecules, chemokines, and chemokine receptors.

**Figure 4 biomedicines-11-02765-f004:**
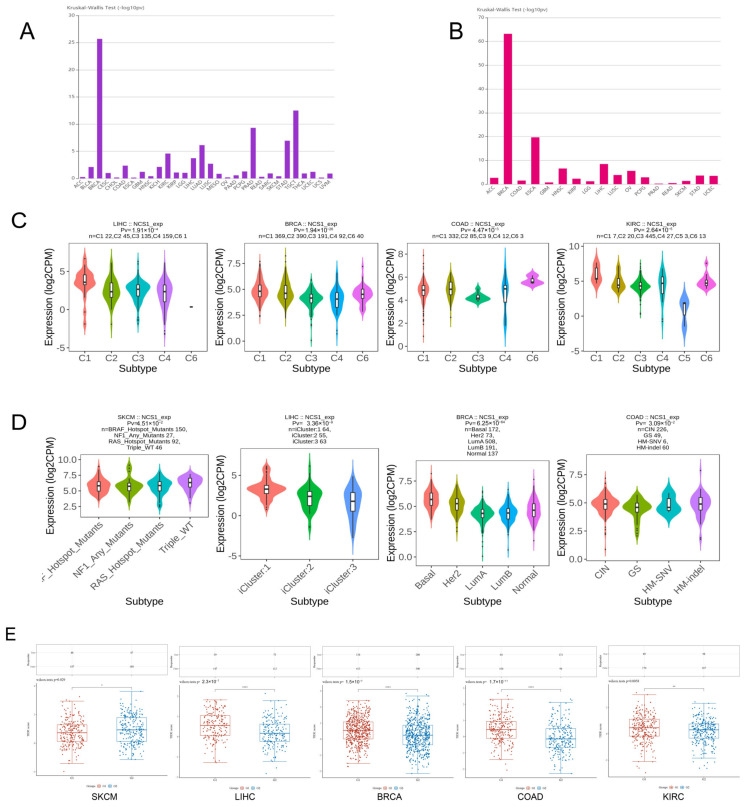
Correlation of NCS1 with immunotherapy in pan-cancer. (**A**) NCS1 predicts different immune subtypes in pan-cancer. (**B**) NCS1 predicts different molecular subtypes in pan-cancer. (**C**) NCS1 can be used to identify different immune subtypes of LIHC, BRCA, COAD, and KIRC. (**D**) NCS1 can be used to identify different molecular subtypes of LIHC, BRCA, COAD, and KIRC. (**E**) NCS1 high expression in SKCM, LIHC, BRCA, COAD, and KIRC shows immune checkpoint resistance correlation. * *p* < 0.05, ** *p* < 0.01, **** *p* < 0.0001.

**Figure 5 biomedicines-11-02765-f005:**
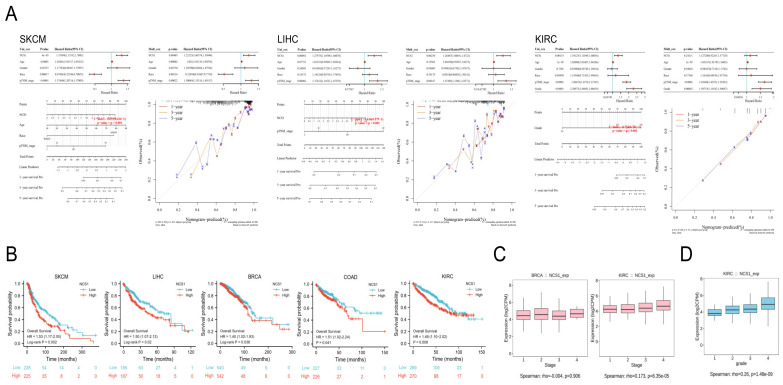
Relationships between NCS1 and prognosis. (**A**) Univariate and multifactorial regression analyses suggested that NCS1 could be a prognostic predictor for SKCM, LIHC, and KIRC. Patients with different NCS1 expression levels showed significant differences in 1-, 3-, and 5-year survival rates; (**B**) KM curves of overall survival of NSC1 in SKCM, LIHC, BRCA, COAD, and KIRC. (**C**) Correlation of NCS1 with pathological stage staging in BRCA and KIRC patients. (**D**) Correlation between NCS1 and tumor grade in KIRC patients.

**Figure 6 biomedicines-11-02765-f006:**
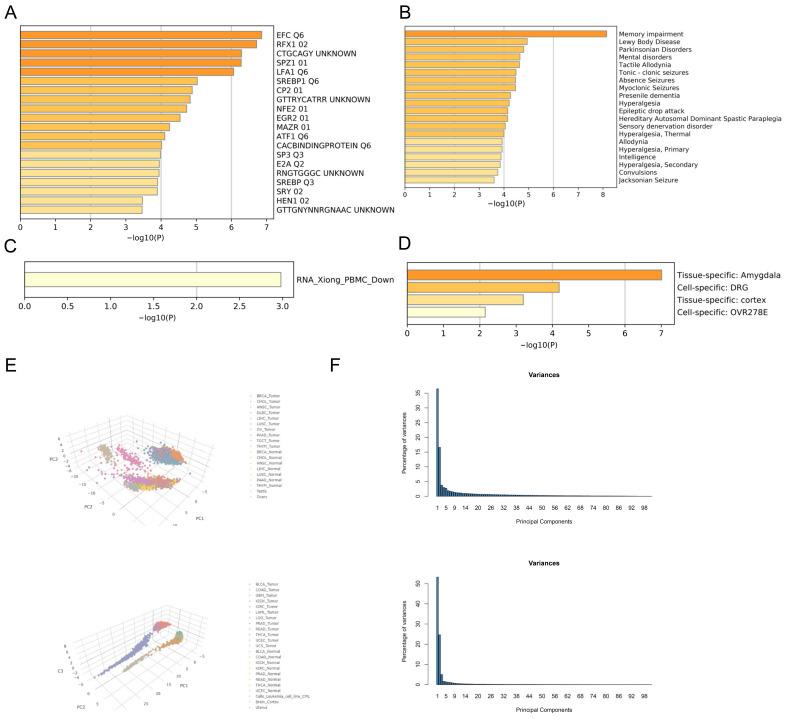
Enrichment and PCA analysis of NCS1 and NCS1-related genes. (**A**–**D**) Enrichment analysis of NCS1 and its similar genes in transcription factor targets, PanGenBase, DisGeNET, and COVID-19. Edge-linked terms with similarity scores > 0.3. (**E**) PCA analysis of NCS1 and NCS1-related genes. (**F**) Variance percentages are based on a PCA analysis of NCS1.

**Figure 7 biomedicines-11-02765-f007:**
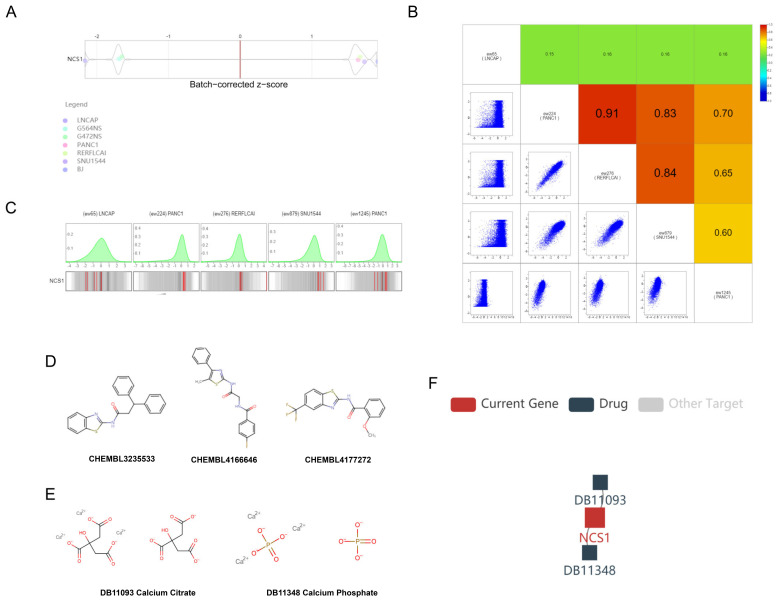
CRISPR screening score of NCS1 and potential compounds targeting NCS1. (**A**) Cell proliferation rate after CRISPR knockout of NCS1. Violin plots show batch-corrected z-scores by the gene for each cell line and screen. Positive Z-scores indicate that gene knockout will promote cell proliferation and survival, whereas negative Z-scores indicate that gene knockout will inhibit it. (**B**,**C**) A gRNA-by-gRNA view shows the knockout efficiency for each guide RNA (red line) with background distribution. The genomic alignment of the gRNA displays the alignment of the gRNA and the z-score of the gRNA effect by color for each screen. (**D**) Drugs targeting NCS1 via the chEMBL databases. (**E**,**F**) Drugs targeting NCS1 via the pharmacogenetics Drugbank and TISIDB database.

## Data Availability

This paper analyzes existing publicly available data. Links to these datasets are given in Materials and Methods. For any additional information needed to reanalyze the data reported in this study, please contact the lead contact.

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
