# Peer review of "The Role of NCS1 in Immunotherapy and Prognosis of Human Cancer"

_biomedicines, 2023, doi:10.3390/biomedicines11102765_

Round 1

Reviewer 1 Report

Wang et al. used machine learning and database mining to assess the role of the protein Neural Calcium Sensor1 (NCS1) in tumor invasion and cell proliferation. Altogether, the work shows that NCS1 has genomic mutations and aberrant DNA methylation in several cancers when compared to normal tissues. The authors’ analysis also points to high NCS1 expression in cancer patients as a marker of to their survival time.

Essential revisions:

While the introduction touches upon major points for the understanding of the relevance of the work and its conclusions, it feels far too succinct at times. The readers would benefit from more context and smoother transitions from one point to the next.

The use of acronyms is so excessive that certain sentences are hard to read or fully understand at first. I suggest authors aim to minimize the use of acronyms and facilitate the reader’s understanding of the narrative. Another effort that can and should be made to improve readability and understanding is to use higher resolution figures, particularly the text on various panels.

While the work reveals some interesting correlations and associations, it is made weak by the lack of follow up in vivo or in vitro experiments to test their hypotheses.

The work is timely and the intriguing findings will present a valuable resource by providing clues from a bioinformatic perspective and highlight the importance of NCS

Minor revisions:

Line 50:  “(…) involves the immune checkpoint (ICB) pathway (…)” The word blockade is missing. ICB should stand for immune checkpoint blockade.

Line 50: TMB stans for tumor mutational burden not tumor mutational load.

Line 51: explain what dMMR stands for.

Line 200 mentions 33 cancer types were surveyed but line 204 mentions 204 cancers instead. Please confirm correct number.

Explain the acronyms LUAD, LUSC, PRAD, UCEC, BLCA, TGCT, ESCA, LIHC, CESC, SARC, BRCA, COAD, SKCM, CHOL, KIRC, THCA, HNSC, LGG, KICH, UVM, and others used to refer to the various cancer types.

Author Response

Response to Reviewer 1 Comments

1. Summary

2. Questions for General Evaluation

Reviewer’s Evaluation

Response and Revisions

Does the introduction provide sufficient background and include all relevant references?

Must be improved

Done, thinks

Are all the cited references relevant to the research?

Yes

Is the research design appropriate?

Can be improved

Done, thinks

Are the methods adequately described?

Can be improved

Done, thinks

Are the results clearly presented?

Can be improved

Done, thinks

Are the conclusions supported by the results?

Can be improved

Done, thinks

  1. Point-by-point response to Comments and Suggestions for Authors

Comments 1: While the introduction touches upon major points for the understanding of the relevance of the work and its conclusions, it feels far too succinct at times. The readers would benefit from more context and smoother transitions from one point to the next.

Response 1: Thanks for your suggestion, we have revised the introduction so that readers can smoothly understand the background, purpose and significance of the study.

Comments 2: The use of acronyms is so excessive that certain sentences are hard to read or fully understand at first. I suggest authors aim to minimize the use of acronyms and facilitate the reader’s understanding of the narrative. Another effort that can and should be made to improve readability and understanding is to use higher resolution figures, particularly the text on various panels.

Response 2: Thinks, Abbreviation has been defined and we have changed figures to improve the picture quality.

Comments 3: While the work reveals some interesting correlations and associations, it is made weak by the lack of follow up in vivo or in vitro experiments to test their hypotheses. The work is timely and the intriguing findings will present a valuable resource by providing clues from a bioinformatic perspective and highlight the importance of NCS.

Response 3: Thanks for your comments, we further highlight the limitations of the study.

Comments 4: Line 50:  “(…) involves the immune checkpoint (ICB) pathway (…)” The word blockade is missing. ICB should stand for immune checkpoint blockade.

Response 4: Done, thinks

Comments 5: Line 50: TMB stans for tumor mutational burden not tumor mutational load.

Response 5: Thank you very much for you reminding. Abbreviation has been corrected.

Comments 6: Line 51: explain what dMMR stands for.

Response 6: Done, thinks.

Comments 7: Line 200 mentions 33 cancer types were surveyed but line 204 mentions 32 cancers instead. Please confirm correct number.

Response 7: Done, thinks

Comments 8: Explain the acronyms LUAD, LUSC, PRAD, UCEC, BLCA, TGCT, ESCA, LIHC, CESC, SARC, BRCA, COAD, SKCM, CHOL, KIRC, TH CA, HNSC, LGG, KICH, UVM, and others used to refer to the various cancer types

Response 8: Done, thinks

Reviewer 2 Report

1. Very small and completely unreadable inscriptions on the pictures, it is impossible to see the details. The drawings should be divided into several smaller ones, but the quality should be improved.

2. In Fig. 1E, is BRCA the same as in Fig. 1F?

3. There are a lot of abbreviations without explanation, including in the pictures.

Overall, a very interesting article, the material is presented consistently and I have no questions regarding the essence of the research. However, the design needs to be corrected.

Author Response

Response to Reviewer 2 Comments

1. Summary

2. Questions for General Evaluation

Reviewer’s Evaluation

Response and Revisions

Does the introduction provide sufficient background and include all relevant references?

Yes

Are all the cited references relevant to the research?

Yes

Is the research design appropriate?

Yes

Are the methods adequately described?

Yes

Are the results clearly presented?

Can be improved

Done, thinks

Are the conclusions supported by the results?

Yes

3. Point-by-point response to Comments and Suggestions for Authors

Comments 1: Very small and completely unreadable inscriptions on the pictures, it is impossible to see the details. The drawings should be divided into several smaller ones, but the quality should be improved.

Response 1: Thank you very much for you reminding. We have changed figures to improve the picture quality.

Comments 2: In Fig. 1E, is BRCA the same as in Fig. 1F?

Response 2: The sample sources of the two pictures are different, and we have described them in more detail in the legend.

Comments 3: There are a lot of abbreviations without explanation, including in the pictures.

Response 3: Thank you very much for you reminding. Abbreviation has been defined and corrected.

Comments 4: Overall, a very interesting article, the material is presented consistently and I have no questions regarding the essence of the research. However, the design needs to be corrected.

Response 4: Done, thinks.

Reviewer 3 Report

The article represents an analysis of NCS1 in the cancer database. It is a bioinformatic well-designed analysis in relationship to the different roles of the Ca binding protein and cancer and part of the immune response. The article can be published as is.

Minor grammatical mistakes

Author Response

Response to Reviewer 3 Comments

1. Summary

2. Questions for General Evaluation

Reviewer’s Evaluation

Response and Revisions

Does the introduction provide sufficient background and include all relevant references?

Can be improved

Done, thinks

Are all the cited references relevant to the research?

Can be improved

Done, thinks

Is the research design appropriate?

Can be improved

Done, thinks

Are the methods adequately described?

Can be improved

Done, thinks

Are the results clearly presented?

Can be improved

Done, thinks

Are the conclusions supported by the results?

Can be improved

Done, thinks

3. Point-by-point response to Comments and Suggestions for Authors

Comments 1: The article represents an analysis of NCS1 in the cancer database. It is a bioinformatic well-designed analysis in relationship to the different roles of the Ca binding protein and cancer and part of the immune response. The article can be published as is.

Response 1: Thank you very much for taking the time to review this manuscript and your positive comments.

Comments 2: Minor grammatical mistakes.

Response 2: The manuscript has been checked thoroughly again and some grammar errors were corrected.